

# Inconsistencies among secondary sources of Chukar Partridge (*Alectoris chukar*) introductions to the United States

Michael P. Moulton[1], Wendell P. Cropper Jr[2] and Andrew J. Broz[1]

[1] Department of Wildlife Ecology and Conservation, University of Florida, Gainesville, FL, USA
[2] School of Forest Resources and Conservation, University of Florida, Gainesville, FL, USA

## ABSTRACT

The propagule pressure hypothesis asserts that the number of individuals released is the key determinant of whether an introduction will succeed or not. It remains to be shown whether propagule pressure is more important than either species-level or site-level factors in determining the fate of an introduction. Studies claiming to show that propagule pressure is the primary determinant of introduction success must assume that the historical record as reported by secondary sources is complete and accurate. Here, examine a widely introduced game bird, the Chukar (*Alectoris chukar*), to the USA. We compare the records reported by two secondary sources (*Long, 1981*; *Lever, 1987*) to those in a primary source (*Christensen, 1970*) and to a recent study by *Sol et al. (2012)*. Numerous inconsistencies exist in the records reported by *Sol et al. (2012)*, *Long (1981)* and *Lever (1987)* when compared to the primary record of *Christensen (1970)*. As reported by *Christensen (1970)*, very large numbers of Chukars were released unsuccessfully in some states. Our results strongly imply that factors other than sheer numbers are more important. Site-to-site differences are the most likely explanation for the variation in success.

## INTRODUCTION

In attempting to identify the processes that deter or promote establishment of introduced bird populations, several empirical studies have concluded that propagule pressure, meaning the total number of individuals of a species released in some place, is the principal determining factor (e.g., *Newsome & Noble, 1986*; *Veltman, Nee & Crawley, 1996*; *Duncan, 1997*; *Green, 1997*; *Cassey et al., 2004*; *Lockwood, Cassey & Blackburn, 2005*; *Sol et al., 2012*). Although this conclusion has been repeatedly criticized (*Moulton et al., 2010*; *Moulton, Cropper & Avery, 2011*; *Moulton et al., 2012*; *Moulton, Cropper & Avery, 2012*; *Moulton, Cropper & Avery, 2013*; *Moulton & Cropper, 2014a*; *Moulton & Cropper, 2014b*; *Moulton & Cropper, 2015*), and recent studies have emphasized the importance of species-level characteristics over propagule pressure (e.g., *Sol et al., 2012*; *Cassey, Prowse & Blackburn, 2014*), some have persisted in touting its primary importance (e.g., *Blackburn, Lockwood & Cassey, 2015*; *Blackburn et al., 2015*).

Corresponding author
Michael P. Moulton,
moultonm@ufl.edu

At the same time, site-level factors have largely been ignored by proponents of propagule pressure, despite numerous studies that have shown their importance in bird introductions (e.g., *Gullion, 1965*; *Diamond & Veitch, 1981*; *Griffith et al., 1989*; *Moulton & Pimm, 1983*; *Moulton & Pimm, 1987*; *Lockwood, Moulton & Anderson, 1993*; *Lockwood & Moulton, 1994*; *Smallwood, 1994*; *Case, 1996*; *Gamarra et al., 2005*; *Moulton & Cropper, 2014b*; *Allen et al., 2015*).

A principal basis for the propagule pressure hypothesis, as applied to birds, has been compilations of historical records such as those by *Thomson (1922)*, *Phillips (1928)*, *Long (1981)*, *Lever (1987)* and *Lever (2005)*. In relying on such secondary sources, studies that claim to support propagule pressure make two assumptions: first that the chronicle of introductions presented in these sources is complete and accurate; and second that the principal, if not sole, motivation behind the introductions was the establishment of self-sustaining populations. A corollary to this second assumption is that introductions would end once it was perceived that the species was established. We show that for Chukar (*Alectoris chukar*) introductions to the USA these assumptions are unmet, and we provide evidence that introduction outcomes in Chukars are likely to be mostly influenced by factors other than numbers released.

Our initial motivation for conducting this study came from the observation that the compilations of *Long (1981)* and *Lever (1987)* often were quite different from that of *Christensen (1970)*, although both cited *Christensen (1970)* in their treatments of the Chukar. *Long (1981)* referred to the species as *Alectoris graeca* but makes it clear that the subspecies involved in the USA were in fact Chukars (Asian origin) and not Rock Partridges (European origin). *Lever (1987)* noted that 'Greek Chukars' released in California were likely Rock Partridges. *Christensen (1970)* discussed the difference in nomenclature referring to North American introductions as *Alectoris chukar*, following the work of *Watson (1962a)* and *Watson (1962b)*. *Lever (1987)* also noted that the species was *Alectoris chukar*, and suggested that the so-called 'Greek Chukars' presented to the state of California were actually Rock Partridges (*Alectoris graeca*).

Historical compilations of bird introductions have often (see above) been used to assess some factors believed to be associated with successful introductions. It is, at least implicitly, assumed that the historical records are either accurate, or that the errors do not significantly bias these analyses. It is difficult to know how complete multi-decade old records actually are, but it is possible to assess the consistency of the major compilations and of the published analyses that have relied on these sources.

## METHODS AND MATERIALS

To illustrate the hazards in depending on secondary sources, we analyzed historical records of introductions of the Chukar to the United States as reported in two major secondary sources: *Long (1981)* and *Lever (1987)*. We then compare the compilations in these two references to the records reported by *Christensen (1970)* and then we show how they compare to the records used in a recent study (*Sol et al., 2012*). *Christensen (1970)* based his compilation on two separate surveys using questionnaires sent to state wildlife agencies

once in the early 1950s and again in the late 1960s. As such, we assume it is the more accurate reflection of the true record of Chukar introductions in the USA.

The Chukar has a vast range throughout Asia (*Watson, 1962a*), and was once considered a subspecies of the Rock Partridge (*Alectoris graeca*), which occurs in Europe. *Watson (1962a)* and *Watson (1962b)* showed that subtle but consistent morphological differences exist between adjacent populations of *A. graeca* and *A. chukar* in extreme Eastern Europe. We follow the 4th edition of the Howard and Moore Checklist of Birds of the World (*Dickinson & Remsen, 2013*), which also treats the two as distinct species.

We compiled lists of introduction records per state as reported by *Long (1981)* and *Lever (1987)*. We then compared these lists to *Christensen (1970)* and *Christensen (1996)*. We compared the number of individuals released in the states for which all three references reported a total number of individuals released. We transformed the total numbers by calculating their common logarithms and then compared these values using a generalized linear mixed model with state (location) of the introduction as a random factor and the three references as a fixed effect. We used the SAS Glimmix procedure (*SAS, 2009*) for our analyses.

We then compare *Christensen*'s (*1970*) list to the records used in the recent study of introductions by *Sol et al. (2012)* and show their degree of reliance on the work of *Long (1981)* and *Lever (1987)*, but not on the seemingly more complete work of *Christensen (1970)*.

## RESULTS

*Bump (1951)* claimed that Chukars had likely been released in every one of the 48 states in the US (Alaska and Hawaii did not become states until 1959) but none of the historical references (*Long, 1981*; *Lever, 1987*; *Christensen, 1970*) listed releases for all 48 states. *Christensen (1970)* and *Christensen (1996)* reported Chukar releases to 40 of the conterminous 48 states (he also noted introductions to Hawaii and Alaska) and listed the total number of individuals released in 35 states (Fig. 1). For five other states (Florida, Louisiana, Michigan, Mississippi, and Rhode Island) respondents reported to *Christensen (1970)* only that a "few" individuals had been released (Table 1). *Long (1981)* reported introductions of Chukars to just 22 states, but only listed propagule information for 16 states. *Lever (1987)* listed releases of Chukars to 30 states, but only reported propagule information for 18 states.

Although *Long (1981)* and *Lever (1987)* both cited *Christensen (1970)*, neither followed his compilation very closely. The reasons that *Long (1981)* and *Lever (1987)* excluded data for so many of the states listed by *Christensen (1970)* are unknown. Moreover, regarding the 15 states for which all three references listed propagule information, *Long (1981)* reported the same number listed by *Christensen (1970)*, for only one state (Missouri) and *Lever (1987)* did not report the same number as *Christensen (1970)* for any state.

*Long (1981)* and *Lever (1987)*, both reported numbers for New York, although *Christensen (1970)* did not. Likely this is due at least in part to *Christensen*'s (*1970*) report being based on wildlife agency surveys and apparently does not include any private

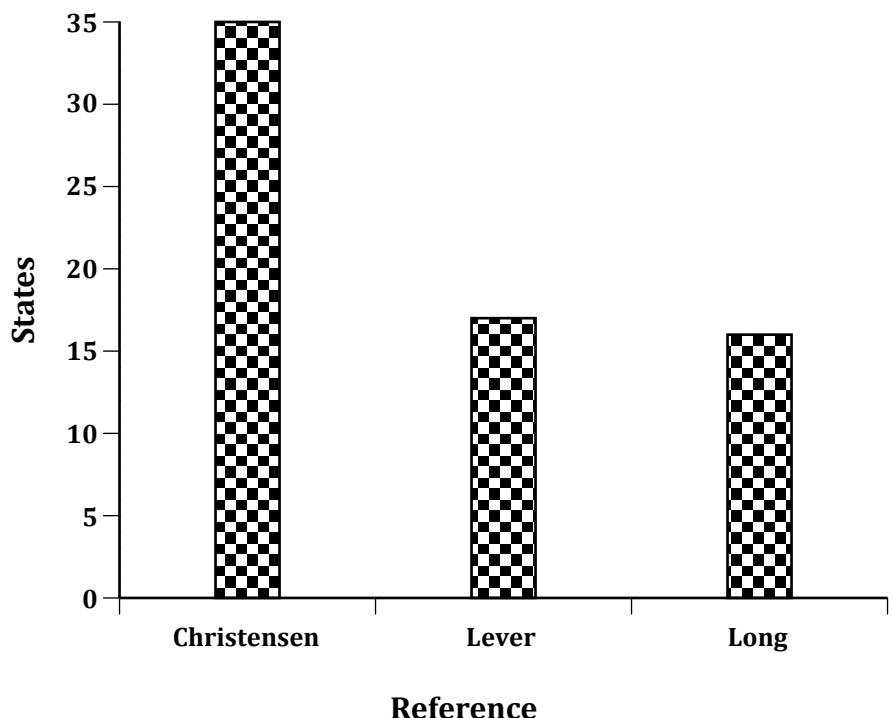

**Figure 1** Number of states reporting total numbers of Chukars released: *Christensen (1970)*, *Lever (1987)* and *Long (1981)*.

releases. *Lever (1987)* also reported numbers for Nebraska and Utah, as did *Christensen (1970)*, but not *Long (1981)*.

In our mixed linear model the logarithms of the numbers of individuals released across the three references and 15 states, with state of introduction as a random effect and reference as a fixed effect, differed significantly in a Type III test ($df$. 2, 20; $F = 4.94$; $p = 0.014$). Clearly, most of the variation in numbers released was due to the higher numbers *Christensen (1970)* reported.

Thus, for unknown reasons, *Long (1981)* and *Lever (1987)* included only about half the states, and significantly fewer individuals than *Christensen (1970)*. We emphasize that none of these references was compiled for the purpose of testing the propagule pressure hypothesis. Nevertheless, we must conclude that results of any studies involving the Chukar that relied heavily on either *Long (1981)* or *Lever (1987)* would likely be based on incomplete and inaccurate information and therefore are suspect.

Studies that presumably include Chukar releases to the USA (e.g., *Cassey et al., 2004*) do not always make their data available. One exception to this is the recent study (*Sol et al., 2012*), which involved a global analysis aimed at disentangling the effects of species-level characters on introduction success in birds. *Sol et al. (2012)* claim to have updated the database used by *Cassey et al. (2004)*.

We were able to match 38 of 40 records of Chukars reported by *Sol et al. (2012)*, using their propagule sizes and ID numbers, to reports by *Long (1981)* or *Lever (1987)* for 16 (or 17) states in the USA (Table 2). *Sol et al. (2012)* did not specify individual states in their

Table 1 **Chukar releases according to Christensen (Ch 1954, Ch 1970),** *Lever (1987)* **and** *Long (1981)*. A question mark indicates that the state was mentioned by the source but no propagule information was available. Chukars are considered established in the ten states in italics: Chukars were considered established in 1954 in the 4 italicized states marked with an asterisk.

| State | Ch 1954 | Ch 1970 | Lever 1987 | Long 1981 | Sol et al., 2012 | FGIP |
|---|---|---|---|---|---|---|
| Alabama[a] | 720 | 720 | ? | ? | . | . |
| *Arizona* | 9,866 | 11,737 | 1,133 | 1,133 | 1,133 | 534 |
| *California** | 44,554 | 55,000 | 75,173 | 39,186 | 14,287 | 11,837 |
| *Colorado* | 10,433 | 24,080 | 8,000 | 9,000 | 9,000 | . |
| Connecticut | 100s | 1,500 | . | . | . | . |
| Florida | Few | Few | ? | . | . | . |
| Georgia | . | . | ? | . | . | . |
| *Idaho** | 8,581 | 25,710 | 28,000 | 28,000 | 25,000 | . |
| Illinois | 9,000 | 9,000 | ? | . | . | . |
| Indiana | . | 7,500 | . | . | . | . |
| Iowa | 1,847 | 1,847 | . | . | . | . |
| Kansas | 7,879 | 7,879 | ? | ? | . | . |
| Kentucky | 15,00 | 5,480 | ? | . | . | . |
| Louisiana | Few | Few | . | . | . | . |
| Maryland | . | . | ? | . | . | . |
| Massachusetts | Few | 500 | ? | . | . | . |
| Michigan | Few | Few | ? | ? | . | . |
| Minnesota | 85,000 | 85,000 | 84,414 | 84,414 | 84,414 | . |
| Mississippi | Few | Few | . | . | . | . |
| Missouri | 1,838 | 1,838 | 1,900 | 1,838 | 1,900 | . |
| *Montana* | 3,629 | 7,854 | 5,365 | 5,365 | 5,365 | . |
| Nebraska | 14,750 | 28,142 | 27,842 | ? | 27,842 | 2,6748 |
| *Nevada** | 6,399 | 13,655 | 5,339 | 6,739 | 5,000 | . |
| New Hampshire | 130 | 130 | . | . | . | . |
| New Mexico | 4,943 | 31,000 | 16,621 | 7,700 | . | 16,471 |
| New York | | | <600 | <600 | 175[b] | |
| North Carolina | 449 | 449 | . | . | . | . |
| North Dakota | 2,300 | 5,600 | ? | . | . | . |
| Ohio | 20 | 20 | . | . | . | . |
| Oklahoma | 1,000s | 1,000s | . | . | . | . |
| *Oregon* | 19,898 | 113,675 | 76,000 | 76,000 | 76,000 | . |
| Pennsylvania | 2,377 | 2,377 | 2,021 | 2,021 | 2,021 | . |
| Rhode Island | . | Few | . | . | . | . |
| South Carolina | Few | 200+ | . | . | . | . |
| South Dakota | 1,459 | 1,831 | 1,368 | 1,368 | 1,368 | 75 |
| Tennessee | 5,824 | 5,824 | ? | ? | . | . |
| Texas | . | 703 | ? | . | . | . |
| *Utah* | 8,666 | 185,911 | 458 | ? | 515 | 73,360 |
| Virginia | 100 | 100 | . | . | . | . |
| *Washington** | 7,041 | 50,920 | 64,996 | 5,841 | 5,841 | 59,155[c] |

Table 1 (*continued*)

| State | Ch 1954 | Ch 1970 | Lever 1987 | Long 1981 | Sol et al., 2012 | FGIP |
|---|---|---|---|---|---|---|
| West Virginia | 4,420 | 4,429 | . | . | . | . |
| Wisconsin | 43,013 | 43,013 | 17,550 | 17,550 | 17,550 | . |
| *Wyoming* | 14,000 | 60,000 | 17,455 | 53,455 | 17,455 | . |
| States | 37 | 40 | 30 | 22 | 17 | 7 |
| Records | 37 | 40 | 69 | 50 | 65 | 154 |
| Individuals | 320,636 | 793,424 | 451,794 | 446,788 | 294,866 | 188,180 |

**Notes.**

[a] These could have been Rock Partridges. *Imhof (1976)* listed "Chukars" in one part of his book and "Rock Partridges" in another, and as *Alectoris graeca* in both places. Moreover he listed the origin of the birds as "southeastern Europe," and did not include the species in a previous publication on birds new to Alabama (*Imhof, 1958*).

[b] Includes by assumption (see text) one unidentified report as being from the state of New York, possibly one for Nebraska (Table 2) and excludes a release attributable to Alaska.

[c] Of these releases, 51,247 occurred between 1970 and 1978 (*Banks, 1981*).

records, but we surmise that they included multiple releases to Arizona (2), California (8), and Utah (14), and single releases (sums) for 13 (or 14—see New York discussion below) others.

*Sol et al. (2012)* listed an unsuccessful record of a propagule size of 175 (Sol et al. ID # - 61), but neither *Long (1981)* nor *Lever (1987)* listed a propagule of this size. It is possible that this represents a conflation of the record *Long (1981)* and *Lever (1987)* listed for Delaware County, New York where 25–150 individuals were released yearly between 1936 and 1939. As shown in Table 2, this record in *Sol et al. (2012)* falls exactly between values and ID numbers we matched to *Lever (1987)* for Missouri (1900—Sol et al. ID # 60) and Pennsylvania (2021—Sol et al. ID # 62). If this record is actually for New York it would represent the fourteenth state as noted above.

*Sol et al. (2012)* also listed two unsuccessful releases of 17 individuals each. One of these possibly refers to 17 individuals released in Alaska (*Lever, 1987*) but the other is uncertain. *Lever (1987)* listed releases to 17 *counties* in Nebraska of 27,842, and it is possible that *Sol et al. (2012)* in the course of updating the data inadvertently included this as a separate release.

We summed multiple releases for Arizona, California and Utah listed by *Sol et al. (2012)* to make their records comparable to the work of *Christensen (1970)*, *Long (1981)* and *Lever (1987)* (Table 3). In a separate mixed model again with state of introduction a random effect and log number of individuals released, we observed a highly significant difference in log number after controlling the random effect of state in the Type III test of fixed effects ($F_{3,45} = 5.88$; $p > F = 0.002$).

We further compared subsets of the sources using two orthogonal contrasts. First, we compared the numbers that *Christensen (1970)* reported per state to those reported by the combination of *Long (1981)*, *Lever (1987)* and *Sol et al. (2012)*. In this contrast we observed a significant difference ($t = 16.60$; $p > t = 0.0002$; $df = 45$). Next we compared the combination of *Long (1981)* and *Lever (1987)* versus *Sol et al. (2012)*, and here the contrast was not significant ($t = 1.01$; $p > t = -0.32$; $df = 45$).

**Table 2  Presumed sources for *Sol et al. (2012)* records.**

| ID | Fate | Prop | State | Lever | Long | Fate |
|---|---|---|---|---|---|---|
| 81 | 1 | 333 | AZ | 1 | 1 | S |
| 3204 | 1 | 800 | AZ | 1 | 1 | S |
| 53 | 1 | 4,600 | CA | 1 | 1 | S |
| 3197 | 1 | 423 | CA | . | 1 | S |
| 3198 | 1 | 444 | CA | . | 1 | S |
| 3199 | 1 | 440 | CA | . | 1 | S |
| 3200 | 1 | 440 | CA | . | 1 | S |
| 3201 | 1 | 440 | CA | . | 1 | S |
| 3202 | 1 | 7,000 | CA | 1 | 1 | S |
| 3203 | 1 | 500 | CA | . | 1 | S |
| 3205 | 1 | 9,000 | CO | 1 | .5 | S |
| 82 | 1 | 25,000 | ID | 1 | 1 | S |
| 59 | 0 | 84,414 | MN | 1 | 1 | F |
| 60 | 0 | 1,900 | MO | 1 | .5 | F |
| 771 | 1 | 5,365 | MT | 1 | 1 | S |
| 1897 | 0 | 27,842 | NE | 1 | ? | F |
| 84 | 1 | 5,000 | NV | 2 | 2 | S |
| 61 | 0 | 175 | NY? | 2 | 2 | F |
| 475 | 1 | 76,000 | OR | 1 | 1 | S |
| 62 | 0 | 2,021 | PA | 1 | 1 | F |
| 1898 | 1 | 1,368 | SD | 1 | 1 | S |
| 88 | 0 | 50 | UT | 1 | . | F |
| 85 | 0 | 13 | UT | 1 | . | F |
| 86 | 0 | 23 | UT | 1 | . | F |
| 87 | 0 | 50 | UT | 1 | . | F |
| 90 | 0 | 41[a] | UT? | 2 | . | F |
| 91 | 0 | 28 | UT | 1 | . | F |
| 92 | 0 | 15 | UT | 1 | . | F |
| 93 | 0 | 15 | UT | 1 | . | F |
| 94 | 0 | 38 | UT | 1 | . | F |
| 95 | 0 | 100 | UT | 1 | . | F |
| 96 | 0 | 8 | UT | 1 | . | F |
| 98 | 0 | 8 | UT | 1 | . | F |
| 97 | 0 | 50 | UT | 1 | . | F |
| 99 | 0 | 76 | UT | 1 | . | F |
| 1587 | 1 | 5,841 | WA | 2 | . | S |
| 467 | 0 | 17,550 | WI | 1 | 1 | F |
| 100 | 1 | 17,455 | WY | 1 | . | S |

**Notes.**

[a] ID 90 of *Sol et al. (2012)* might be a typographical error, as *Lever (1987)* listed a release of 46 to Utah.
ID refers to the ID number in *Sol et al. (2012)*; Fate, 1 successful, 0, unsuccessful; Prop, propagule size as listed by *Sol et al. (2012)*. Lever and Long refer to the presence of the record in those two references (*Long, 1981*; *Lever, 1987*): .5, fewer listed by the reference; 1, identical number listed; 2, additional releases to the state were listed by the reference. The Fates are those *Sol et al. (2012)* reported (S, Successful; F, Failed).

**Table 3 Chukar release summary by various sources: Ch70,** *Christensen (1970)***; Le87,** *Lever (1987)***; Lo81,** *Long (1981)***; Sol,** *Sol et al. (2012)***.**

| State | Ch70 | Le87 | Lo81 | Sol |
|---|---|---|---|---|
| Nevada | 13,655 | 5,339 | 6,739 | 5,000 |
| California | 55,000 | 75,173 | 39,186 | 14,287 |
| Colorado | 24,080 | 8,000 | 9,000 | 9,000 |
| Wyoming | 60,000 | 17,455 | 53,455 | 17,455 |
| Idaho | 25,710 | 28,000 | 28,000 | 25,000 |
| Washington | 50,920 | 64,996 | 5,841 | 5,841 |
| Arizona | 11,737 | 1,133 | 1,133 | 1,133 |
| South Dakota | 1,831 | 1,368 | 1,368 | 1,368 |
| Missouri | 1,838 | 1,900 | 1,838 | 1,900 |
| Pennsylvania | 2,377 | 2,021 | 2,021 | 2,021 |
| Montana | 7,854 | 5,365 | 5,365 | 5,365 |
| Wisconsin | 43,013 | 17,550 | 17,550 | 17,550 |
| Oregon | 113,675 | 76,000 | 76,000 | 76,000 |
| Minnesota | 85,000 | 84,414 | 84,414 | 84,414 |
| New Mexico | 31,000 | 16,621 | 7,700 | . |
| Utah | 185,911 | 458 | . | 515 |
| Nebraska | 2,8142 | 2,7842 | . | 27,842 |
| New York | . | <600 | <600 | 175? |

## DISCUSSION

The first assumption of the propagule pressure hypothesis mentioned above was that the historical record was complete and accurate. Whereas there might be more complete and accurate records that are not generally well known, secondary sources such as *Long (1981)* and *Lever (1987)* are seemingly incomplete and likely inaccurate. Studies such as *Sol et al. (2012)* and presumably *Cassey et al. (2004)* apparently relied heavily on the reports in *Lever (1987)* and *Long (1981)* but as we have shown here neither author completely or accurately reflected the introduction data presented by *Christensen (1970)*. Thus, for Chukar introductions to the USA we have shown that the record as presented by *Long (1981)* and *Lever (1987)* appears to be incomplete and inaccurate.

The second assumption is that all the individuals that were introduced were necessary for establishment. Chukars currently have self-sustaining populations in ten western states (see Table 1). In four of these states (California, Idaho, Nevada, and Washington) Chukars were considered established in 1954 (*Christensen, 1954*); in the other six states (Arizona, Colorado, Montana, Oregon, Utah, and Wyoming) the status was considered uncertain, doubtful (Arizona) or hopeful (Utah, Oregon). However, additional individuals were released in all ten states between 1954 and 1970 (*Christensen, 1970*), strongly suggesting that establishment of wild Chukar populations was not the only goal. If propagule pressure was assessed as an essential factor by the professionals introducing these birds, we might expect the six states where the status was uncertain to release larger numbers after 1954 than the four states where the Chukar was considered established. As indicated in Table 1,

*Christensen (1954)* considered Chukars to be established in four states (California, Idaho, Nevada, and Washington). However, by 1970 additional individuals were released in all four states (California—10,446; Idaho—17,129; Nevada—7,256; Washington—43,879). Thus, even in those states where the population of Chukars was considered established, releases continued. In fact, introductions continued for years after *Christensen*'s (*1970*) report. Thus, *Banks (1981)* further reported that in the state of Washington where the Chukar was considered established by 1954, more than 51,000 Chukars were released between 1970 and 1978.

As noted by *Duncan, Blackburn & Sol (2003)* three levels of factors could influence introduction outcome in birds: species-level; event-level; and site-level. As we focus here solely on *Alectoris chukar*, we can ignore the possibility that species-level differences could explain differences in introduction outcomes. Could other event-level characteristics be responsible? Possible event-level factors, other than propagule pressure, include characteristics of the releases themselves. Some studies (e.g., *Veltman, Nee & Crawley, 1996*; *Sol et al., 2012*), include releases of diverse sets of species that likely were made under differing circumstances, and with different goals. For example, the conditions involved in releases of species introduced for biological control likely differed from those of species released for aesthetic reasons. Such diverse releases likely were made by groups or individuals with different goals. We note that the Chukars were introduced chiefly, if not exclusively, to provide recreational hunting opportunities. The numbers of individuals released in the different states, reported by *Christensen (1970)* came from questionnaires sent to state game and fish departments throughout the USA. The Chukar releases *Christensen (1970)* reported were presumably all made by state sponsored professional wildlife scientists and so it is unlikely that differences in introduction outcomes across the states could simply reflect differences in the levels of competence among personnel in the different states. Despite the seeming homogeneity in Chukar introduction practices, in several states very large numbers of Chukars were unsuccessfully released. For example, 85,000 individuals were released into Minnesota, more than 43,000 into Wisconsin, and more than 28,000 in Nebraska, only to fail.

The results here strongly imply that factors other than sheer numbers, and characteristics of the release events determined the outcome of Chukar introductions. Thus, the logical explanation is that site-level factors such as climate or habitat characteristics (*Gullion, 1965*) were of greater importance than sheer numbers in determining the outcome of Chukar introductions. Indeed, the only states with successful Chukar populations are states that straddle or are west of the continental divide. These states share certain environmental characteristics: all are more arid and mountainous than states where Chukars failed (*Johnsgard, 1988*; *Christensen, 1996*).

### Funding

The authors received no funding for this work.

## Competing Interests

The authors declare there are no competing interests.

## Author Contributions

- Michael P. Moulton conceived and designed the experiments, performed the experiments, analyzed the data, wrote the paper, prepared figures and/or tables, reviewed drafts of the paper.
- Wendell P. Cropper Jr conceived and designed the experiments, performed the experiments, analyzed the data, wrote the paper, reviewed drafts of the paper.
- Andrew J. Broz performed the experiments, reviewed drafts of the paper.

## Data Availability

This work did not generate any raw data.

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
