# Peer review of "Inconsistencies among secondary sources of Chukar Partridge (Alectoris chukar) introductions to the United States"

_PeerJ, doi:10.7717/peerj.1447_

## Round 0.1 · original submission · Major Revisions

I will be happy to see this again if you can address the reviewers' comments. My personal view is that this is an important piece of the discussion about the role of propagule pressure. The manuscript does need a careful restructuring however and must address the concerns raised.

·

Basic reporting

No comments

Experimental design

No comments

Validity of the findings

No comments

Additional comments

Great premise — better clarity on factors influencing success/failure of reintroductions is definitely needed, and this makes the valuable point that conclusions drawn on historical data must first validate that historical data (or at least discuss the context within which it exists). Would very much like to hear more about the data sources — where did the records come from? Why were subsequent reports so much lower than Christensen’s early data?

86 - 96 — does this belong in introduction, rather than methods?

105 - 107 — I'm a bit confused here; were you showing the reliance of Sol’s records on Long & Lever? Or is this referring to the lists you generated from the Long, Lever, and Christensen datasets?

134 — Should be ‘most of’

153 — Discussion of possible New York data point could be a new paragraph, if you like.

183 — Needs comma between ‘accurate’ and ‘secondary’

187 — Are we presuming that Christensen’s (1970) record is the most accurate? If so -- why? These examinations seem to suggest that everything must be held in comparison to Christensen’s work, but without knowing more about his methods or hearing a justification for why his numbers should be presumed to be the most accurate, it’s unclear why that should be so. Could a sentence be added to tell us where Christensen's data came from (was this person a key decision maker in the chukar reintroductions)?

197 - 199 — I follow the conclusion here, but would like a bit more context for chukar reintroductions. Were there economic drivers (e.g. sport hunting industries that were more well-established in certain states) that should make us expect releases to be driven by something other than a desire for self-sustaining reintroduced population?

204 - 208 — Confused what this is in reference to?

211 — Needs comma between ‘released’ and ‘only’

212 — Needs comma after ‘Indeed’

215 — Definitely agree that you have thoroughly demonstrated with the chukar reintro history that propagule pressure is *not* the leading contributor to success/failure, but not certain that I follow the jump to site-level factors? Could more citations and explanation in reference to the predominance of site selection be added?

Reviewer 2 ·

Basic reporting

The basic reporting is sound, although, as I note below, the reader would benefit from more information about the key publications that are analyzed in this study.

Experimental design

The authors of this manuscript have two goals in mind: (1) to examine whether sources of information, in particular secondary sources, on the locations and numbers of chukar introductions in the USA are used in a consistent and accurate manner by researchers exploring the factors behind the success or failure of animal introductions. (These data have been used to examine whether propagule pressure is a better predictor of the success of an introduction than are site-level factors); (2) to explore whether site-level factors are more important than propagule pressure (harkening back to the underlying question the inspired the first goal).

Goal 1: Sources of information. Their analysis reveals differences in chukar introduction data between the two widely-cited secondary sources (Long 1981, Lever 1987), as well as inconsistences between Long (1981) and Lever (1987) and what is presumably a primary source, Christensen (1970), although the authors of this manuscript do not explicitly call it a primary source. This is to some degree important, because the tone of the manuscript suggests that deviations from Christensen (1970) are problematic without telling us why Christensen should be regarded as something of a gold standard for chukar introductions. Is there a reason to believe that Long or Lever relied exclusively on Christensen, in which case any deviations from his records would indeed suggest sloppiness? Nonetheless, that Long, Lever, and Christensen don’t all agree is clearly established by the authors.

The authors then point out that the data on chukars used by Sol et al. (2012), which Sol et al. claim were borrowed and updated from Cassey et al. (2004), don’t match Long, Lever, or especially Christensen. Frankly, at this point, I began to feel as though I was caught up in a version of that old childhood game “telephone,” in which a string kids try to pass the same message along verbally and discover that the last kid to hear the message hears something utterly different from what the first kid actually said. The logic of this part of the paper escapes me: why should we expect Sol to agree with Long, Lever, or Christensen when Sol has utilized and updated a database from Cassey, and no one knows the source of Cassey’s data? Perhaps it’s just the fact that they all are inconsistent, but, if so, the authors need to make this point more explicitly.

Propagule pressure versus site-level factors: This really comes up only in the last paragraph of the manuscript, where the authors note that thousands of chukars have been pumped into many states with little evidence of establishment; the only places where the chukars are taken hold are the drier western states, ergo site-level factors outweigh propagule pressure. I think the authors are right about this, at least with respect to chukars in the USA. But what they have shown here is hardly a rigorous test of the two competing hypotheses, and this discussion does not belong in this manuscript (and especially not in the abstract).

Validity of the findings

Please see my comments above. In particular, I do not think this manuscript adequately tests the propagule-pressure vs. site-level factors debate; therefore, I do not believe they can or should include findings on this issue (although, I hasten to add, I actually think they are right about their conclusions--I just don't think they have done a proper examination of the issue).

Additional comments

See under Experimental Design.

---

## Round 0.2 · accepted · Accept

Dear Mike:

Previously you had two reviewers, one who liked it and one who had a lot of concerns. The former likes your revision, not surprisingly, but the other reviewer is now unavailable.

I've gone over the revision and I feel that this makes an important contribution. While I do not need to see this again, I do want you to make some changes.

Now, the main text of the paper starts with the main idea: Propagule pressure. It then points out that its controversial and that there are alternative hypotheses. Then in the third paragraph, you argue that the propagule pressure hypothesis makes two assumptions and that those are unmet. Finally, you notice that the data sources fail to do the necessary job.

I think this is excellent structure to your ideas! But, alas, I do not like your abstract for the simple reason it does not follow your entirely crisp, compelling introduction. In contrast, your abstract essentially starts with the data being poor, never articulates alternative hypotheses, doesn't tell us what the propagule pressure hypothesis predicts, nor why we should care about data quality. So, please rewrite the abstract so that it follows the logic of your introduction.

·

Basic reporting

No comments

Experimental design

No comments

Validity of the findings

No comments

Additional comments

Really interesting and much clarified work!